# Modeling of Rapid Response Characteristics of Piezoelectric Actuators for Ultra-Precision Machining

**DOI:** 10.3390/ma16062272

**Published:** 2023-03-11

**Authors:** Bowen Zhong, Zhan Liao, Xi Zhang, Ziqi Jin, Lining Sun

**Affiliations:** The College of Mechanical and Electrical Engineering, Soochow University, Suzhou 215137, China

**Keywords:** piezoelectric actuators, dynamic model, experiment of bouncing steel ball with piezoelectric actuator

## Abstract

Piezoelectric actuators are characterized by high positioning accuracy, high stiffness and a fast response and are widely used in ultra-precision machining technologies such as fast tool servo technology and ultrasonic machining. The rapid response characteristics of piezoelectric actuators often determine the overall quality of machining. However, there has been little research on the fast response characteristics of piezoelectric actuators, and this knowledge gap will lead to low precision and poor quality of the final machining. The fast response characteristics of a piezoelectric actuator were studied in this work. Firstly, the piezoelectric actuator was divided into a no-load state and a load state according to the working state. A fast response analysis and output characteristic analysis were carried out, the corresponding dynamic model was established, and then the model was simulated. Finally, an experimental system was established to verify the dynamic model of the piezoelectric actuator’s fast response by conducting an experiment in which the piezoelectric actuator bounces a steel ball. The experimental results verify the correctness of the model and show that the greater the cross-sectional area and height of the piezoelectric actuator, the higher the bouncing height of the ball, and the better the dynamic performance of the piezoelectric actuator. It is believed that this study has guiding significance for the application of the dynamic characteristics of piezoelectric actuators in the machining field.

## 1. Introduction

A piezoelectric actuator is a type of actuator that uses the inverse piezoelectric effect to convert electrical energy into mechanical motion [1,2,3]. In addition, piezoelectric actuators are widely used in various ultra-precision machining technologies due to their remarkable advantages, such as a fast dynamic response, high resolution and large output force [4,5,6]. In particular, the high dynamic characteristics of a piezoelectric actuator make it suitable for ultrasonic machining [7], fast tool servo technology (FTS) [8], ultra-precision cutting technology [9] and even machining vibration detection [10]. More and more applications for the high dynamics of piezoelectric actuators have attracted the attention of researchers [11,12,13].

For example, Yang, Y. et al. [14] developed a slender turning tool with an aspect ratio of 7 by using a shock absorber equipped with piezoelectric ceramics. Considering the double damping of piezoelectric ceramics and the rubber gasket, the equation of motion was established. Experimental results show that cutting tools with shock absorbers can achieve an 80.1% reduction. Hu, CJ et al. [15] designed a micro-blanking device that takes advantage of the fact that the output displacement of piezoelectric ceramics varies with the input voltage. By using deform software, micro internal gear punching was simulated and analyzed, and the ideal internal gear parts were obtained. Compared with other equipment for processing such parts, the device is characterized by good processing performance, low cost and high control precision. M.A.A. Viera et al. [16] describe the performance evaluation of a low-cost thin-disk piezoelectric diaphragm (PZT) for surface integrity monitoring during grinding. Here, PZT refers to a piezoelectric material. It is a solid solution of PbZrO_3_ and PbTiO_3_. The evaluation results show a clear relationship between thin-disk PZT diaphragms and conventional acoustic emission (AE) sensors for grinding surface integrity monitoring, demonstrating the effectiveness and reliability of innovative sensing devices for current applications. Zhu, Z.H. et al. [17] reported high-performance triaxial FTS with hybrid electromagnetic–piezoelectric actuation and a hybrid parallel–serial-kinematic structure. To generate the planar motion as well as to carry the piezoactuated vertical motion, a novel axis-symmetric linearized reluctance actuator (AS-LRA) was proposed. Having good accordance with the design targets, the prototype demonstrated over 50 μm and 15 μm travel with very slight cross-talk for the planar and vertical motions, respectively. However, the above studies are based on structural innovation after building relevant dynamic models to improve performance. The research in this paper starts from the power source, namely, the piezoelectric actuator, and presents the corresponding dynamic model by studying its dynamic characteristics in order to provide guidance for its design and application in the machining field, improve the performance of the piezoelectric actuator and improve the overall machining quality.

The purpose of this paper is to provide theoretical guidance for the application of piezoelectric actuators. In order to improve the overall quality of ultra-precision machining, firstly, piezoelectric actuators were divided into a no-load state and a load state according to their working states. The fast response characteristics and output characteristics were analyzed, and the corresponding dynamic model was established. Then, according to the obtained dynamic model, the differential method was used to simulate it, and an experimental method in which the piezoelectric actuator bounces a steel ball was proposed. By comparing the simulated values with the experimental values, the influence of the piezoelectric actuator’s cross-sectional area, height and external load on the maximum velocity was verified, and the correctness of the proposed dynamic model was verified. The research results have a guiding significance for the application of piezoelectric actuators in the field of ultra-precision machining.

## 2. Rapid Response Analysis

Figure 1 presents a schematic diagram of the polarization process of the piezoelectric actuator. After polarization treatment, the piezoelectric actuator has a fixed polarization direction, and thus, it has anisotropy. When an external electric field is applied, the polarized piezoelectric actuator exhibits an inverse piezoelectric effect and can be used as an actuator element.

Since the piezoelectric actuator used in the research and experiment in this paper is the longitudinal length stretching vibration mode of the cylinder, that is, the −33 vibration mode, the constitutive equation of piezoelectric materials in the −33 mode is obtained according to Reference [18], and the mechanical equation is as follows:(1)$$S_{3}=s_{33}^{E}T_{3}+d_{33}E_{3}$$

In the above, $S_{3}\;$ is the strain tensor, $s_{33}^{E}{}^{}$ is the elastic compliance matrix when subjected to a constant electrical field, $T_{3}$ is the stress tensor, $d_{33}$ is the longitudinal (output displacement direction) piezoelectric strain coefficient of the piezoelectric actuator, and $E_{3}$ is the electric field vector.

The fast response characteristic of the piezoelectric actuator is the dynamic inertia force characteristic of the piezoelectric actuator without a load. For the dynamic application of the piezoelectric actuator, inertia force and tension are two alternating forces, and the inertia force is much larger than the tension. However, the tension has a great influence on the piezoelectric actuator, so even if the piezoelectric actuator has no external load, the influence of the dynamic force should still be considered. Therefore, in view of the load-free inertial driving characteristics of the piezoelectric actuator, combined with the constitutive equation of the piezoelectric actuator, an experiment based on the piezoelectric actuator’s self-bouncing ability under the excitation of the driving voltage was designed on the basis of the action of the inertia force. The motion state of the piezoelectric actuator in different driving stages was analyzed, and the influence of the excitation voltage on the jump time speed and jump height was clarified.

The fast response characteristic model of the piezoelectric actuator is shown in Figure 2. Figure 2a shows the excitation voltage diagram of the piezoelectric actuator at each stage (where U on the vertical axis represents the voltage applied to both ends of the piezoelectric actuator), Figure 2b shows the state of the piezoelectric actuator at each stage during voltage excitation, and Figure 2c shows the force analysis diagram of the piezoelectric actuator at each stage. The motion process can be introduced in four stages, as described below.

Stage A (initial state): The piezoelectric actuator remains at its original length since there is no driving voltage excitation. At this time, the piezoelectric actuator is affected by the ground’s supporting force $F_{N}$ and gravity mg.

Stage B (before ejection): The piezoelectric actuator is extended toward both ends to generate output displacement due to the excitation of the stage B voltage. Because inertia force and output displacement are two output characteristics occurring at the same time, the actual output performance of the piezoelectric actuator will generate an inertia force at both ends. So, in the end, the piezoelectric actuator contacts the ground, but because of the inertia force of the ground support, $F_{N}^{\prime }$ > $F_{N}$ with an acceleration of a.

Point C (at ejection): When the driving voltage reaches point C, the piezoelectric actuator is at the boundary point between contact with and departure from the ground. At this point, to face upward, the force of the piezoelectric actuator tends to zero, and its acceleration at this time is of $a^{\prime }$.

Stage D (after ejection): In this stage, the piezoelectric actuator starts to elongate and jump. Then, the piezoelectric actuator jumps to the highest point at the initial speed and returns to the ground in a state of free fall. Under the action of the excitation voltage, the piezoelectric actuator will recover to its original length at some point.

Through the analysis of the motion state and force of the piezoelectric actuator, it can be seen that the moment at which bouncing occurs is when the piezoelectric actuator is decelerating; that is, its excitation voltage is in the state of negative acceleration. The corresponding excitation voltage in the step-up stage of the piezoelectric actuator should be divided into two stages: positive and negative acceleration. Therefore, the input voltage in the step-up stage of the piezoelectric actuator can be set as:(2)$$V\left(t\right)=\{\;\begin{array}{cc} A\left(e^{t}-1\right) & 0⩽t⩽t_{0}\\ B\ln t+c & t_{0}⩽t⩽t_{e} \end{array}$$
where A, B and c are constants $\left(V\right)$; $t_{0}$ is the boundary time of boosted acceleration and deceleration; and $t_{e}$ is the moment when the voltage step of the piezoelectric driver ends.

Under the condition that the input voltage of the piezoelectric actuator is known (the waveform combining the exponential function and logarithmic function is used in this paper to simulate the voltage curve), and according to the extension principle of the piezoelectric actuator, the expressions for the extension quantity, velocity and acceleration of the piezoelectric actuator can be obtained as follows:(3)$$u\left(t\right)=nd_{33}V\left(t\right)=\{\;\begin{array}{cc} nd_{33}\left(e^{t}-1\right)\cdot A & 0⩽t⩽t_{0}\\ nd_{33}\left(B\ln\;t+c\right) & t_{0}⩽t⩽t_{e} \end{array}$$
(4)$$v\left(t\right)=nd_{33}V{\left(t\right)}^{\prime }=\{\;\begin{array}{cc} nd_{33}e^{t}\cdot A & 0⩽t⩽t_{0}\\ \frac{nd_{33}B}{t} & t_{0}⩽t⩽t_{e} \end{array}$$
(5)$$a\left(t\right)=nd_{33}V{\left(t\right)}^{\prime \prime }=\{\;\begin{array}{cc} nd_{33}e^{t}\cdot A & 0⩽t⩽t_{0}\\ -\frac{nd_{33}B}{t^{2}} & t_{0}⩽t⩽t_{e} \end{array}$$
where n is the number of layers of the piezoelectric driver. Since the set voltage is continuous at the demarcation point, and because of the extension of the piezoelectric actuator, the speed does not have a law causing a sudden change at the demarcation point, and the excitation voltage of the piezoelectric actuator can be obtained. The value of the extension of the piezoelectric actuator and the speed at the demarcation point are equal, namely:(6)$$\left\{\begin{array}{c} A\left(e^{t_{0}}-1\right)=Bln\;t_{0}+c\\ Ae^{t_{0}}=\frac{B}{t_{0}}\\ Bln\;t_{e}+c=V_{max} \end{array}\right.$$

In the formula, the first two formulas are valid under the condition $t=t_{0}$, and the third formula is valid under the condition $t=t_{e}$, where $V_{max}$ is the maximum excitation voltage $\left(V\right)$ of the piezoelectric actuator. According to Equation (6), the expression of the piezoelectric actuator excitation voltage parameter B about $t_{0}$ can be obtained as follows:(7)$$B=\frac{V_{max}t_{0}e^{t_{0}}}{e^{t_{0}}-1+t_{0}e^{t_{0}}ln\;\frac{t_{e}}{t_{0}}}$$

The piezoelectric actuator began to bounce, and according to the analysis, the upward force of the piezoelectric actuator tended to zero, and it had an acceleration of $a^{\prime }$ at this time. The force in this state can be expressed as:(8)$$m_{eff}a^{\prime }=-mg$$

In the formula, $m_{eff}$ is the equivalent mass of the piezoelectric actuator. g is $10\;m/s^{2}$, and the acceleration of the piezoelectric actuator is −30 $m/s^{2}$, that is, $-B/t_{2}=-30$; that is, when $t=\sqrt{B/30}$, the piezoelectric actuator starts to bounce, and the expressions of velocity and elongation at this moment can be obtained, respectively, as:(9)$$v{\left(t\right)}_{jump}=\sqrt{30nd_{33}B}$$
(10)$$u{\left(t\right)}_{jump}=\sqrt{30B}nd_{33}$$

Then, the actual bouncing of the piezoelectric actuator can be expressed as (when the piezoelectric actuator is in the extended $\left(a\right)$ and contracted $\left(b\right)$ state, respectively):(11)$$h{\left(t\right)}_{jump}=\left\{\begin{array}{c} \frac{nd_{33}B}{15}-\sqrt{30B}nd_{33}\;\;\;\;\;\;\;\;\;\;\;\;\;\;\;\;\;\;\;\;\;\;t=\sqrt{\frac{2nd_{33}B}{150}}\;\;\left(a\right)\\ \frac{nd_{33}B}{15}-\left(\Delta l_{max}+\sqrt{30B}nd_{33}\right)\;\;t=\sqrt{\frac{2nd_{33}B}{150}}\;\;\left(b\right) \end{array}\right.$$
where $\Delta l_{max}$ represents the maximum elongation of the piezoelectric actuator (μm). The actual jump height of the piezoelectric actuator can be obtained by setting the value of $t_{0}$ and selecting the model of the piezoelectric actuator. The actual jump height of the piezoelectric actuator is related to the cut-off point $t_{0}$, the maximum value of the excitation voltage $V_{max}$ and the number of ceramic pieces n in the piezoelectric actuator. Therefore, the influence of these three parameters on the bouncing height of the piezoelectric actuator will be discussed on the basis of the simulation and experimental results in the following sections.

## 3. Output Characteristic Analysis

The stress, strain and electric field of the piezoelectric brake in Formula (1) are replaced by their basic mechanical property formula, which can be the output displacement of the N-layer piezoelectric driver:(12)$$ku\left(t\right)-nkd_{33}V\left(t\right)=F$$
where k is a constant and represents the stiffness of the piezoelectric actuator, $u\left(t\right)$ is the displacement of the piezoelectric actuator, $d_{33}$ is the longitudinal (output displacement direction) piezoelectric strain coefficient of the piezoelectric actuator, $V\left(t\right)$ is the voltage applied across the piezoelectric actuator, and F is the load applied to the piezoelectric actuator.

In this study, an experiment in which the piezoelectric actuator bounces a steel ball was designed to further analyze Formula (12). In the experiment, one end of the piezoelectric actuator should be fixed on the table, and the steel ball should be placed on the other. Under the excitation of the driving voltage, the piezoelectric actuator extended and then returned to its original length. During this period, the steel ball moved with the actuator for a period of time and then separated from the actuator. Subsequently, the steel ball continued to move upward at a different speed. A schematic diagram of the piezoelectric actuator bouncing a steel ball in one cycle is shown in Figure 3.

According to Figure 3, the working process of the piezoelectric actuator bouncing the steel ball can be divided into several steps. They are described as follows.

At time 0, the piezoelectric actuator is de-energized, and then the piezoelectric actuator and the steel ball are in contact, which can be called the “initial state”.

The piezoelectric actuator is energized. Due to the elongation of the piezoelectric actuator caused by the voltage, the steel ball has an upward velocity with the piezoelectric actuator’s extension, but during this period, the steel ball is not separated from the piezoelectric actuator. We can call this period “common movement”.

Under the action of voltage, the piezoelectric actuator continues to stretch; meanwhile, the steel ball and the piezoelectric actuator are separated at the moment when the acceleration of both is −g (acceleration of gravity). We call the separation moment the “separation point”.

After the steel ball is separated from the piezoelectric actuator, the actuator continues to stretch under voltage excitation. The steel ball moves upward by a certain distance at the speed of separation and then moves downward.

According to the schematic diagram of the steel ball being bounced by the piezoelectric actuator, the steel ball on the piezoelectric actuator is analyzed for further analysis of Equation (12). The stress analysis of the steel ball bounced by the piezoelectric actuator in the experiment is shown in Figure 4.

According to Newton’s third law of action and reaction, the magnitude of the output force of the piezoelectric actuator is equal to the load applied to it. So, F and $F_{output}$ are two forces of equal magnitude. H in Figure 4 is the jump height of the steel ball. In the experiment, the speed at which the result is converted into the separation time of the steel ball can be recorded. Moreover, the output force of the piezoelectric actuator can be written as $F_{output}=m_{eff}u\left(t\right)\prime \prime$. Considering the direction of the force, Equation (12) can be converted into:(13)$$ku\left(t\right)-nkd_{33}V\left(t\right)=-m_{eff}u{\left(t\right)}^{\prime \prime }.\;$$
where $m_{eff}$ is the equivalent mass, and it can be expressed as $m_{eff}=m_{T}+\frac{1}{3}m$. Using the differential method, the acceleration of the equivalent mass can be decomposed into $u{\left(t\right)}^{\prime \prime }=\frac{u\left(t\right)-2u\left(t-1\right)+u\left(t-2\right)}{∆t^{2}}$, which leads to Formula (13), expressed as
(14)$$\left(k+\frac{m_{eff}}{∆t^{2}}\right)u\left(t\right)-\frac{2m_{eff}}{∆t^{2}}u\left(t-1\right)+\frac{m_{eff}}{∆t^{2}}u\left(t-2\right)=nkd_{33}V\left(t\right)$$

In the above equations, ∆t is the time required for the elongation to change from $u\left(t-1\right)$ to $u\left(t\right)$; the equation for $V\left(t\right)$ can be acquired by an oscilloscope, and $u\left(t\right)$ can be solved using the differential method in Matlab. Since the speeds of the piezoelectric actuator and steel ball are the same before their separation, the speed of the piezoelectric actuator and steel ball at the separation moment can be calculated by $u\left(t\right)$.

## 4. Comparison of Experiment and Simulation

Figure 5 presents the experimental system diagram of the piezoelectric actuator bouncing the steel ball. The system is constituted by a personal computer (PC), signal generator, power amplifier, oscilloscope, height ruler, camera, piezoelectric actuators and steel balls. Figure 6 shows the established experimental apparatus of the piezoelectric actuator bouncing the steel ball. Table 1 shows the specific models of various experimental equipment. The PC-controlled signal generator provides the ideal waveform. The waveform voltage is amplified by the power amplifier and then transmitted to the piezoelectric actuator. Under the excitation voltage, the piezoelectric actuator lengthens or shortens. Therefore, the steel ball on the piezoelectric actuator bounces. Meanwhile, the camera is used to record the jump height of the steel ball. Since the steel ball will move upward at the moment after it is separated from the piezoelectric actuator, the speed at the separation moment can be calculated by measuring the jump height of the steel ball. At last, all of the data are gathered and processed by the PC. In order to obtain satisfactory experimental data, the whole system was fixed on a vibration-isolated optical table, and all the experiments were conducted at a temperature of about 25 °C.

In order to verify the correctness of the above model, an experiment was performed in which piezoelectric actuators were used to bounce steel balls. The model was verified by measuring the speed at which the steel ball and the piezoelectric actuator separated. Various parameters of all piezoelectric actuators used in this experiment are shown in Table 2.

The first set of experiments involved the same piezoelectric actuators bouncing steel balls with different masses. In other words, $m_{eff}$ in Equation (14) was changed. The purpose of this method is to change the equivalent mass to verify the correctness of the dynamic model of the piezoelectric actuator. The parameters of the first set of experiments are shown in Table 3.

As can be seen from the results shown in Figure 7, the simulation curve is consistent with the calculated curve. Furthermore, the first set of experiments prove that the dynamic model of the piezoelectric actuator’s rapid response is correct.

The second set of experiments involved piezoelectric actuators with different cross-sectional areas of steel balls bouncing at the same height. In other words, k and $m_{eff}$ in Equation (14) were changed to verify the correctness of the dynamic model of the piezoelectric actuator. For this analysis, two groups of comparative experiments were designed, and their parameters are shown in Table 4, and experimental results are shown in Figure 8.

In the second set of experiments, the change in the actuator capacitance is caused by the change in the piezoelectric actuator’s stiffness, which affects the change in the input voltage $V\left(t\right)$. Therefore, the experiment above does not show obvious rules, but it can verify that the dynamic model of the piezoelectric actuator’s rapid response is correct.

The third set of experiments involved piezoelectric actuators with different bounce heights of steel balls with the same cross-sectional area. In other words, $m_{eff}$ and k in Equation (14) and the height of the actuator were changed to verify the correctness of the dynamic model of the piezoelectric actuator’s rapid response. For the third series of experiments, two groups of comparative experiments were designed, and their parameters are shown in Table 5, and experimental results are shown in Figure 9.

In the third set of experiments, the height of the actuator was changed, resulting in a change in the capacitance of the piezoelectric actuator, which in turn affected the input voltage $V\left(t\right)$ of the piezoelectric actuator. Therefore, the experiment above does not show obvious rules, but it can verify that the dynamic model of the piezoelectric actuator’s rapid response is correct.

Comparing the above three sets of experimental calculation results and simulation results proves the correctness of the established dynamic model. It can be seen from the experimental calculation results of the separation speed of the steel ball are smaller than the simulation results because of the energy lost during transmission and the error in the readings. Therefore, the rapid response dynamic model established in this paper is correct. This provides a prerequisite for the application of the dynamic model in the machining field.

## 5. Conclusions

In order to improve the overall quality of ultra-precision machining and provide theoretical guidance for the application of piezoelectric actuators, a dynamic model of the rapid response and output characteristics of piezoelectric actuators is established in this paper. We adopted the method of dynamic modeling to try to give an intuitive and clear expression of the dynamic characteristics of the piezoelectric actuator. Firstly, the piezoelectric actuator was divided into a no-load state and a load state according to its working state. The fast response analysis and output characteristic analysis were carried out, and the corresponding dynamic model was established. A series of laws were obtained for the fast response characteristics of the piezoelectric actuator with the cross-sectional area, height and load. An experimental method in which the piezoelectric actuator bounces a steel ball was proposed, and the maximum bouncing velocities of steel balls with different cross-section areas, bouncing heights and masses were studied. The correctness of the dynamic model was verified by comparing the simulated values with the experimental values. The experimental results verify the correctness of the model and show that with the greater the cross-sectional area and height of the piezoelectric actuator, the higher the bouncing height of the ball, and the better the dynamic performance of the piezoelectric actuator. The research results have certain guiding significance for the application of piezoelectric actuators in the field of ultra-precision machining.

## Figures and Tables

**Figure 1 materials-16-02272-f001:**
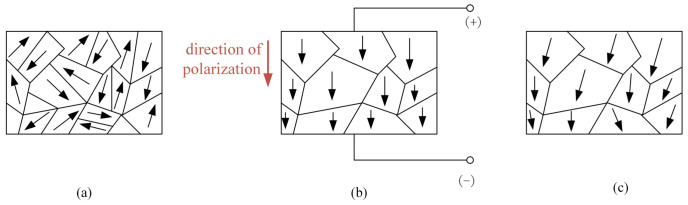
Schematic diagram of polarization process of piezoelectric actuator. (**a**) Before polarization; (**b**) during polarization; (**c**) after the polarization.

**Figure 2 materials-16-02272-f002:**
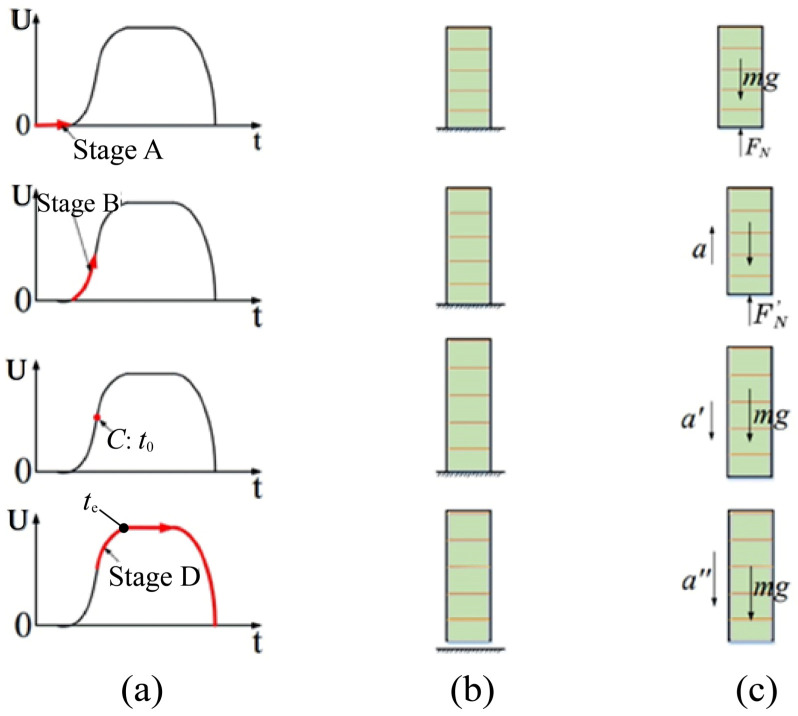
Motion and force analysis of piezoelectric actuator without load. (**a**) Excitation voltage diagram of the piezoelectric actuator; (**b**) State of the piezoelectric actuator; (**c**) Force analysis diagram of the piezoelectric actuator.

**Figure 3 materials-16-02272-f003:**
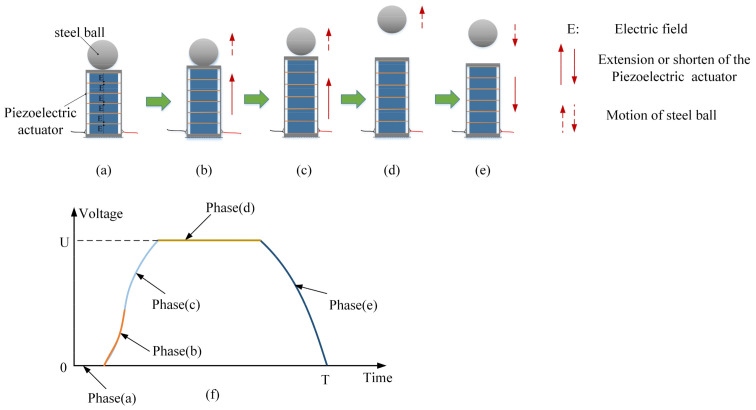
A schematic diagram of the piezoelectric actuator bouncing the steel ball in one cycle. (**a**–**e**) Section diagram of the process of piezoelectric actuator bouncing steel ball; (**f**) Drive voltage of piezoelectric actuator.

**Figure 4 materials-16-02272-f004:**
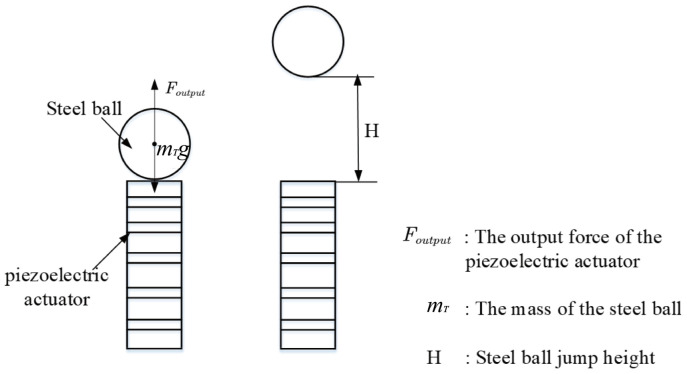
The stress analysis of the steel ball bounced by the piezoelectric actuator in the experiment.

**Figure 5 materials-16-02272-f005:**
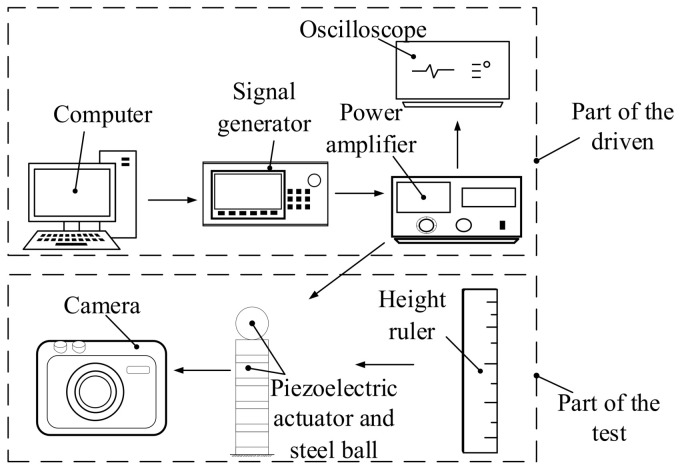
Experimental system diagram of the piezoelectric actuator bouncing the steel ball.

**Figure 6 materials-16-02272-f006:**
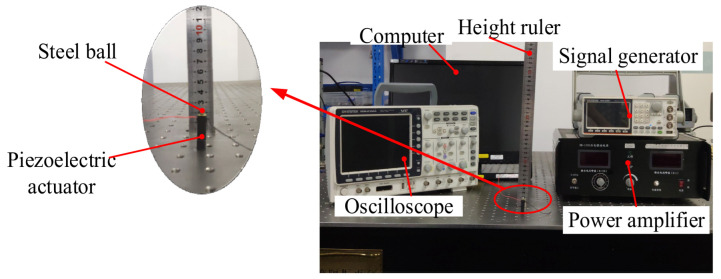
The established experimental apparatus of the piezoelectric actuator bouncing the steel ball.

**Figure 7 materials-16-02272-f007:**
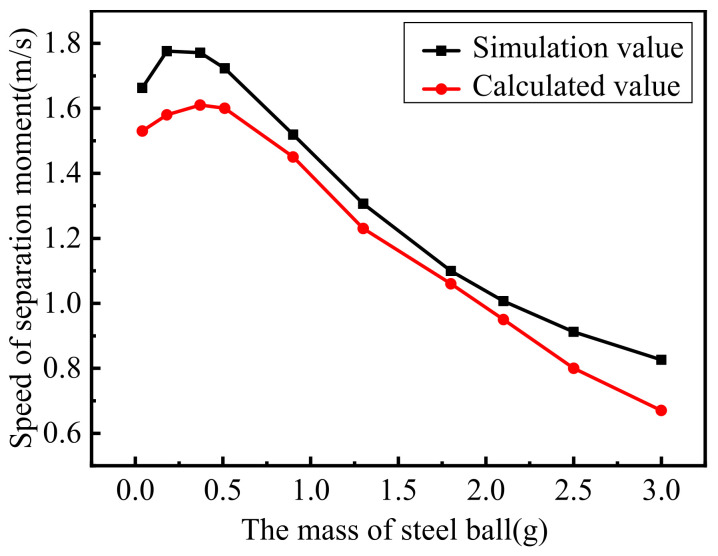
Comparison of experimental and simulated values of the first set of experiments.

**Figure 8 materials-16-02272-f008:**
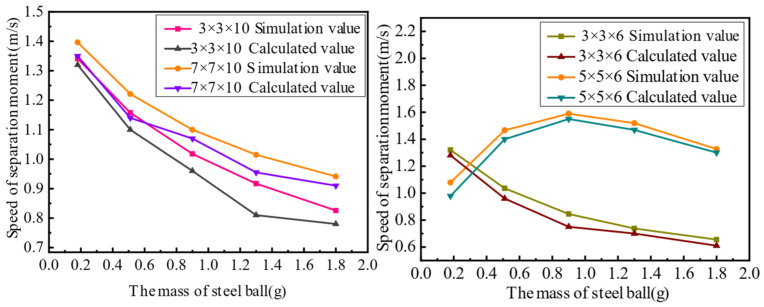
Comparison of simulated and experimental values in the second set of experiments.

**Figure 9 materials-16-02272-f009:**
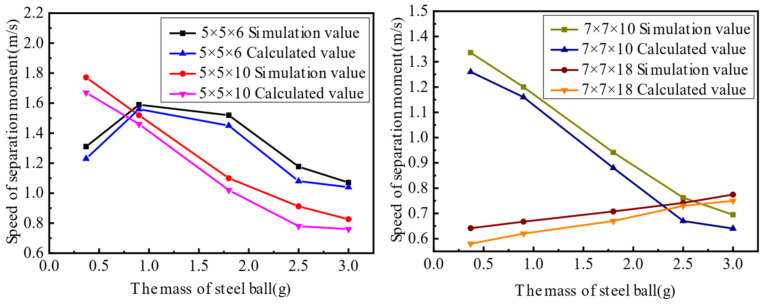
Comparison of simulated and experimental values in the third set of experiments.

**Table 1 materials-16-02272-t001:** Specific models of experimental equipment.

Equipment	Unit Type	Manufacturer
Signal generator	AFG-3051	Guwei Electronics Co., CHN
Piezoelectric ceramic drive power	DR-1501	Nantong Longyi Electronic Technology Co., CHN
Oscilloscope	GDS-2104A	Guwei Electronics Co., CHN
Capacitance gauge	CapaNCDT6300	German Mil Test Co. CHN
High-precision digital display table	2000 Multimeter	Keithley Co., USA
High-speed camera	MV-CA050-10GM	Hikvision Digital Technology Co., CHN

**Table 2 materials-16-02272-t002:** Detailed parameters of various types of piezoelectric actuators used in the experiment.

Dimensions	3 × 3 × 6	3 × 3 × 10	5 × 5 × 6	5 × 5 × 10	7 × 7 × 10	7 × 7 × 18
Capacitance (μF)	0.18	0.3	0.44	0.8	1.6	3
Stroke (μm)	5	10	5	10	10	20
Mass m (g)	0.42	0.69	1.2	1.98	3.9	6.96
Stiffness k (N/μm)	66	33	180	90	180	90
Number of layers n	100	100	100	100	100	100
Voltage $V\left(t\right)$ (V)	≤150	≤150	≤150	≤150	≤150	≤150

**Table 3 materials-16-02272-t003:** Parameter table for the first set of experiments.

Mass of Steel Ball (g) (Piezoelectric Actuator: 5 mm × 5 mm × 10 mm)									
0.04	0.18	0.37	0.51	0.9	1.3	1.8	2.1	2.5	3

**Table 4 materials-16-02272-t004:** Parameter table for the second set of experiments.

Size of Piezoelectric Actuator (mm)	Mass of Steel Ball (g)				
3×3×10	0.18	0.51	0.9	1.3	1.8
7×7×10					
3×3×6					
5×5×6					

**Table 5 materials-16-02272-t005:** Parameter table for the third set of experiments.

Size of Piezoelectric Actuator (mm)	Mass of Steel Ball (g)				
5×5×6	0.37	0.9	1.8	2.5	3
5×5×10					
7×7×10					
7×7×18					

## Data Availability

Data sharing is not applicable to this article. No new data were created or analyzed in this study.

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
