# Peer review of "Modeling of Rapid Response Characteristics of Piezoelectric Actuators for Ultra-Precision Machining"

_materials, 2023, doi:10.3390/ma16062272_

Round 1

Reviewer 1 Report

Please introduce in the text of the paper the replies to the questions, using the track changes tool.

Author Response

Thank you very much for your valuable advice. We believe that your suggestions can make our manuscript more suitable for publication. We have carefully read your suggestions and made comprehensive revisions to the manuscript. Please refer to the document for details.

Reviewer 2 Report

The comments as attached

Author Response

(The authors gave the same response as above.)

Reviewer 3 Report

Uwagi:

·       Line 44: Abbreviation (PZT) should be explianed.

·       Line 46: Abbreviation AE should be explianed.

·       Line 46: Abbreviation (FTS) is already explianed in line 30.

·       Line 51: Abbreviation AS-LRA  should be explianed.

·       Line 72: There should be one word "polarization".

·       In Eq. (1) and (2) confusing is s for two different quantities

·       Line 76-78: Describe the symbols in eq. (1) and (2)  precisely. What does Equation (2) describe?

·       Figure 2: Explain the symbols. Is stage 1 the same as stage A, 2=B, FN=Fn, etc.? Confusing.

·       Equation (3) is from literature or experiment?

·       What is x(t) and n in eq.(4)?

·       Give a reference that you can count v and a, like in eq. (5) and (6).

·       What means h w eq. (12) and how it is related to x?

·       Equation (13) is from literature or experiment? How u(t) is related to x?

·       In Figure 3, consider whether P is the same as the direction of the electric field E. (see Fig. 1)

The descriptions in the Figure 7-9 are not visible well.

Conclusions need to be improved to show significance of content and originality / novelty.

Author Response

(The authors gave the same response as above.)

Reviewer 4 Report

General Comments: This paper presents an alternate method for analysis of response and output characteristics of the piezoelectric actuator, using steel ball as a load. Dynamic characteristics of the investigated piezoelectric actuators are analyzed theoretically as well as experimentally. This work is interesting as the piezoelectric actuator is one of the key equipment in precision engineering fields. There are a few items that need attention. Specific comments and questions needing attention are provided below.

Specific Comments:

1. Page 2, lines 51-52:  a new abbreviation AS-LRA is used in the text – should be explained.

In the text  prototype demonstrated an over 50 a unit of the value 50 should be added.

2. The indices of the parameters used in formulas (1) and (2) should be explained.

3. In Fig.2 the subfigures (a), (b) and (c) should be marked.

4. Page 4, lines 122 – 124: t0 and te should be explained graphically in Fig.2. Please explain te=20µs.

5. Page 4, line 137:  Please explain Vmax  150 V and what actuator model was analyzed.

6. Page 4, lines 142-143: A statement …which is generally one third of the mass of the piezoelectric actuator.,  should be discussed in detail or reference should be presented.

7. In Fig.3 a Phase (e) should be corrected.

8. Page 6, lines 202-204: mass mT should be explained; the differential method used for analysis should be discussed in detail or reference should be presented.

9. Page 7, lines 211-214, 218: information (model, company, country) about the equipment used in experiments should be presented.

10. Page 7, a methodology of the simulated velocity measurement should be explained.

11. Please explain how the slew rate of the power amplifier affect motion response of the piezoelectric actuator.

There are some grammar errors in the manuscript, carefully check and correct them.

Author Response

(The authors gave the same response as above.)

Round 2

Reviewer 2 Report

The authors have fixed it based on comment

Author Response

Thank you for your approval.

Reviewer 3 Report

Remarks and Comments (ver.2):

The authors referred superficially to my earlier remarks.

·       Line 48: Abbreviation (PZT) should be explianed. PZT is the best known piezoelectric material. It is a solid solution of PbZrO3 and PbTiO3 - check in the literature.

·       Your reply: Thank you for your advice, because these two questions come from formula 1 and 2. Please allow me to answer these two questions together. As you said, there is indeed ambiguity in the formula s and s3, and formulas 1 and 2 are both from Chapter 2, Part B of Literature 15. It is negligent for us to only mention s in the following discussion, because s3 in formulas 1 and 2 represents the mechanical constitutive equation of piezoelectric ceramics in the -33 mode. We omitted this point in the paraphrase and have now corrected it in the original text.

My reply: And in the test (line 85) you gave ref. [18]. You correctly mentioned that the piezoelectric actuator has anisotropy. Why in Eq. (1) it's ?33? and you only explain ??. Please specify also E3 as electric field. ?33 is piezoelectric coefficient (thickness mode) - check in the literature. Again this remark: Explain what Equation (2) describes? Do you need equation (2)? This is a direct piezoelectric effect. The reader feels that the authors are not sufficiently familiar with piezoelectricity.

·       Your reply: ... As mentioned in the previous paragraph of Formula 3, the acceleration change rule is obtained after the analysis of the four stages in Fig.2. I think you may have such a question because there are still many problems in the expression of the four stages above.

My reply: What is U in Figure 2? I think t means time. How does U relate to the force FN? Does stage A in Fig. 2 refer to step 1 in the text (stage B to step 2, etc.)?

·       Again this remark: Equation (3) is from literature or experiment? Write it in the text or provide ref.

·       Your reply: Thank you for your careful advice, since these questions all come from x(t) in Formula 4, please allow me to answer them together. x(t) in formula 4 actually represents the elongation of piezoelectric actuator. The original intention here is to prevent it from being confused with V(t) in formula (3) above, but the unity with the later text is ignored. Now it is changed to u(t) in the text. This is also modified. Finally, I am sorry that I did not find h and w in formula 12. I think you may mean ????? and ? , which was indeed not explained clearly in the original text and has been revised in the original text.

My reply: Write it in the text that u(t) in formula 4 represents the elongation of piezoelectric actuator. You can find h in version1 of your manuscript:

What means h w eq. (12) and how it is related to x (u in ver.2 of your manuscript)?

·       Your reply: To calculate formula 5 and 6, the values of ?0 and ?? need to be set first. Here, we take 20 and 30 respectively. So we can figure out the constant B according to formula 8. Formula 8 is as follows:

My reply: It's easier to give ref. or from basic mechanics define velocity and acceleration.

·       Again this remark: Eqution (13) is from literature or experiment? Write it in the text or provide ref.

·       In Figure 3, I propose to insert the electric field E in place of the polarization P in Fig.3.

Author Response

Thank you very much for your suggestions. I believe your comments will help our manuscript to be published smoothly. I have noticed that in the last round of review, I may have misunderstood your meaning for some questions, which led to some questions that I did not reply to you clearly. In this round of review, I have made corresponding replies to these questions.

Reviewer 4 Report

General Comments: This paper presents an alternate method for analysis of response and output characteristics of the piezoelectric actuator, using steel ball as a load. Dynamic characteristics of the investigated piezoelectric actuators are analyzed theoretically as well as experimentally. This work is interesting as the piezoelectric actuator is one of the key equipment in precision engineering fields. There are a few items that need attention. Specific comments and questions needing attention are provided below.

Specific Comments:

1. In Abstract and in Conclusions please explain meaning of an expression: … spring stiffness of the ball

2. Page 2, the text … prototype demonstrated an over 50Hz…should be explained.

3. In Fig.2 the subfigures (a), (b) and (c) should be marked.

4. I would recommend the parameters t0 and te should be shown in Fig.2.

5. Technical characteristics (dimensions, weight, capacitance, stroke...) of the piezoelectric actuators used in this research  should be presented.

6. Page 5, lines 152-153: A statement …which is generally one third of the mass of the piezoelectric actuator.,  should be discussed in detail or reference should be presented.

7. Page 7, line 214: mass mT should be explained.

9. In your experiments a same power amplifier is used to excite different models of the piezoelectric actuators. Please explain how the slew rate of the power amplifier affect motion response of the piezoelectric actuators with different capacitances.

Author Response

(The authors gave the same response as above.)
